# Crispant analysis in zebrafish as a tool for rapid functional screening of disease-causing genes for bone fragility

Sophie Debaenst[1], Tamara Jarayseh[1], Hanna De Saffel[1], Jan Willem Bek[1], Matthieu Boone[2], Ivan Josipovic[2], Pierre Kibleur[2], Ronald Y Kwon[3,4], Paul J Coucke[1], Andy Willaert[1]*

[1]Center for Medical Genetics Ghent, Department of Biomolecular Medicine, Ghent University, Ghent, Belgium; [2]Center for X-ray Tomography, Department of Physics and Astronomy, Ghent University, Ghent, Belgium; [3]Department of Orthopaedics and Sports Medicine, University of Washington, Seattle, United States; [4]Institute for Stem Cell and Regenerative Medicine, University of Washington, Seattle, United States

## eLife Assessment

The paper presents a streamlined new approach for functional validation of genes known to underlie fragile bone disorders in a relatively high throughput, using CRISPR-mediated knockouts and a number of phenotypic assessments in zebrafish. **Convincing** data demonstrate the feasibility and validity of this approach, which presents an **important** tool for rapid functional validation of candidate gene(s) associated with heritable bone diseases identified from genetic studies.

*For correspondence: andy.willaert@ugent.be

Competing interest: The authors declare that no competing interests exist.

**Abstract** Heritable fragile bone disorders (FBDs), ranging from multifactorial to rare monogenic conditions, are characterized by an elevated fracture risk. Validating causative genes and understanding their mechanisms remain challenging. We assessed a semi-high throughput zebrafish screening platform for rapid *in vivo* functional testing of candidate FBD genes. Six genes linked to severe recessive osteogenesis imperfecta (OI) and four associated with bone mineral density (BMD) from genome-wide association studies were analyzed using CRISPR/Cas9-based crispant screening in F0 mosaic founder zebrafish. Next-generation sequencing confirmed high indel efficiency (mean 88%), mimicking stable knock-out models. Skeletal phenotyping at 7, 14, and 90 days post-fertilization (dpf) using microscopy, Alizarin Red S staining, and microCT was performed. Larval crispants showed variable osteoblast and mineralization phenotypes, while adult crispants displayed consistent skeletal defects, including malformed neural and haemal arches, vertebral fractures and fusions, and altered bone volume and density. In addition, *aldh7a1* and *mbtps2* crispants experienced increased mortality due to severe skeletal deformities. RT-qPCR revealed differential expression of osteogenic markers *bglap* and *col1a1a*, highlighting their biomarker potential. Our results establish zebrafish crispant screening as a robust tool for FBD gene validation, combining skeletal and molecular analyses across developmental stages to uncover novel insights into gene functions in bone biology.

## Introduction

Heritable fragile bone disorders (FBD), which are generally associated with increased fracture risk, range from common multifactorial conditions to rare monogenic diseases (*Charoenngam et al., 2023*). They present within a wide phenotypic spectrum where monogenic disorders are generally more

severe and often include skeletal deformities and extra-skeletal abnormalities. Multifactorial FBD, on the other hand, usually present with milder symptoms that are often restricted to bone fragility. Osteoporosis is a common skeletal disease characterized by low bone mass, low bone mineral density (BMD) and increased fracture risk (*Kanis et al., 1994*). It affects about 200 million people around the world (*Pouresmaeili et al., 2018*). In Europe, 5.6% of the population aged over 50 is diagnosed with osteoporosis, making it a highly prevalent disease, especially in the elderly. The risk of osteoporosis is influenced by both environmental factors, such as physical activity, diet preferences, medication use, coexisting diseases, and genetic factors (*Ralston and Uitterlinden, 2010*). Low BMD and fractures are highly heritable risk factors, with heritability values ranging from 50% to 80% (*Richards et al., 2012*). Genome-wide association studies (GWAS) in humans have unraveled hundreds of susceptible loci associated with BMD and other bone traits related to risk of osteoporotic fracture (*Zhu et al., 2021*). GWAS results have great potential for the development of drug targets, the identification of risk factors, and risk prediction for osteoporosis. However, the identification of causal genes at these loci or the mechanisms by which GWAS loci alter bone physiology has faced challenges due to the scarcity of *in vivo* validation tools with sufficient throughput (*Sabik and Farber, 2017*).

In monogenic FBD, on the other hand, gene identification is generally more straightforward. One example of such a disease is osteogenesis imperfecta (OI), a genetically heterogeneous connective tissue disorder that is mainly characterized by bone fragility, skeletal deformities, and other extra skeletal manifestations (*Forlino and Marini, 2016*). Although in the majority of cases, it is caused by autosomal dominant mutations in the collagen type 1 genes, mutations in an increasing number of non-collagen genes causing severe recessive forms of OI, are being identified. These genes encode proteins that are involved in different aspects of bone formation, such as osteoblast differentiation (OSTERIX, WNT1, OASIS, S2P, and SPARC), bone mineralization (PEDF and BRIL) and collagen processing (HSP47, FKBP65, LH2, KDELR2, CRTAP, P3H1, CyPB, and TRIC-B) (*Forlino and Marini, 2016*; *Jovanovic et al., 2022*). Recent sequencing technologies have improved genetic diagnosis, but the investigation of the pathogenesis of the different forms of OI is still a laborious, costly, and time-consuming task. Hence, it is essential to develop efficient and cost-effective methods for *in vivo* functional testing to validate the involvement and mechanistic role of candidate and newly identified causal genes in the pathogenesis of FBD.

Zebrafish are a valuable model organism for studying human diseases, as they share a high degree of genetic similarity with humans. More than 70% of human protein-coding genes and 82% of human disease-related genes have zebrafish orthologs, including most of the genes involved in skeletogenesis (*Howe et al., 2013*). Additionally, zebrafish share many of their skeletal features and ossification mechanisms with mammals (*Dietrich et al., 2021*) and numerous zebrafish models for skeletal diseases have been established and investigated (*Gistelinck et al., 2018*; *Daponte et al., 2023*; *Kague and Karasik, 2022*). These models have some advantages over rodent models, such as longer survival, likely due to the reduced impact of gravity on their fragile skeletons in the aquatic environment, and the availability of transgenic lines marking specific bone cell types combined with the transparency of early life stages allowing for *in vivo* observation of the dynamics of skeletal development (*Dietrich et al., 2021*).

Genome-editing with CRISPR/Cas9 is a fast and easy method to create knockout alleles and stable zebrafish mutant lines (*Uribe-Salazar et al., 2022*). However, obtaining zebrafish lines that are homozygous for the mutant alleles requires two generations of adult animals (>6 months), which limits the efficiency and affordability of genetic screens (*Kroll et al., 2021*). To overcome these drawbacks, phenotyping the first-generation (F0) mosaic founder zebrafish, also known as crispants, is an emerging approach. Adult crispants have a shorter generation time (~3 months) than stable lines (6–9 months), which allows screening more genes in less time and at lower costs and resources. We previously demonstrated the feasibility of this approach by comparing the phenotypes and molecular profiles of a stable knockout line and a crispant for the osteoporosis gene *lrp5*, and found them to be highly similar (*Bek et al., 2020*). In a study by *Watson et al., 2020*, comparison of crispants to homozygous germline mutants for *bmp1a* and *plod2*, two causal OI genes, revealed phenotypic convergence, suggesting that crispants faithfully recapitulate the biology of germline skeletal disease models. Therefore, while zebrafish crispant screens are promising as an economic, fast, and biologically relevant genetic screening approach in vertebrate model systems, a rigorous evaluation of their potential to screen for FBD-associated genes is needed.

In this study, we aimed to conduct a pilot-study to evaluate the feasibility of zebrafish crispant screening as a gene-based approach for rapid *in vivo* functional screening and validation of a set of causal genes for a broad range of FBD. The selected set includes on one hand genes associated with traits related to osteoporosis which is a relatively mild late-onset FBD and on the other hand genes causing recessive forms of OI associated with more severe skeletal phenotypes. An overview of the selected genes with observed mutant phenotypes in human, mice and zebrafish is provided in *Supplementary file 1a*. With this study, we want to provide an experimental platform for medium-sized zebrafish crispant screening of FBD genes.

## Results
### Generation of zebrafish crispants for osteoporosis and OI genes
To identify and prioritize potential causal genes for osteoporosis, we used the GWAS catalog (*Sollis et al., 2023*). We selected the variants associated with bone density and ranked them by their GWAS evidence, i.e., the number of independent publications that reported significant associations. We then identified the genes that harbored these variants and chose the top four candidates with single zebrafish orthologs: *ALDH7A1*, *ESR1*, *DAAM2,* and *SOST*. Furthermore, we selected six genes associated with recessive forms of OI associated with severe skeletal phenotypes, and involved in different aspects of bone formation, including osteoblast differentiation (*CREB3L1*, *MBTPS2, and SPARC*), bone mineralization (*SERPINF1* and *IFITM5*) and collagen transport (*SEC24D*).

For each of the 10 selected genes and 'scrambled' sequence, Alt-R gRNAs (IDT) were designed through a thorough procedure, via the Benchling platform. For each gene, the gRNA with the highest out-of-frame (OOF) efficiency, as predicted by the InDelphi-mESC prediction tool (*Figure 1—figure supplement 1*) was selected for co-injection with Cas9 into one-cell stage embryos. Based on previous studies, we assumed that high indel (insertion/deletion) and OOF efficiencies result in a sufficient reduction or inactivation of the corresponding protein in crispants to induce a skeletal phenotype (*Naert et al., 2020*). DNA from a pool of 1 dpf larvae (n=10) for the different crispants and corresponding sibling controls, was extracted, and subjected to NGS analysis. Sequencing data was analyzed using the Crispresso2 tool (*Clement et al., 2019*) to determine the fraction of reads containing indels, the OOF rate and the number of in-frame deletions consisting of more than 6 base pairs in these pools (*Table 1*).

Across the different crispants the minimal indel efficiency was 71%, while the OOF rate ranged between 49 and 73%. We subsequently determined the proportion of reads containing in-frame deletions of at least six base pairs, resulting in the removal of two or more amino acids (*Table 1*; *Figure 1—figure supplement 1*). Such deletions have a relatively high chance of impairing protein

**Table 1.** InDelphi-mESC prediction, indel, out-of-frame indel and in-frame rates for the different crispants, determined by NGS from pools of 1dpf embryos.

The first four genes are associated with the pathogenesis of osteoporosis, while the last six are linked to osteogenesis imperfecta. The percentage of reads likely to affect the protein function is calculated based on the percentage of reads with out-of-frame indels combined with the percentage of reads with in-frame deletions of equal to or more than 6 basepairs.

|  | aldh7a1 | daam2 | esr1 | sost | creb3l1 | ifitm5 | mbtps2 | sec24d | serpinf1 | sparc |
|---|---|---|---|---|---|---|---|---|---|---|
| % reads with indel | 87% | 96% | 92% | 80% | 71% | 90% | 94% | 85% | 92% | 92% |
| % reads with out-of-frame indel | 66% | 57% | 64% | 64% | 49% | 73% | 73% | 63% | 55% | 72% |
| % reads with in frame indel ≥6 bp | 16% | 24% | 15% | 14% | 11% | 15% | 19% | 12% | 25% | 16% |
| % reads likely affecting protein function | 82% | 81% | 79% | 78% | 60% | 88% | 92% | 75% | 80% | 88% |

function (*Emond et al., 2020*; *Savino et al., 2022*; *Miton and Tokuriki, 2023*). The accumulated percentage of reads with indels that likely cause loss-of-function of the respective protein (OOF % + % in-frame indel ≥6 bp), fall within the range of 60% to 92%, although also reads with short in-frame indels (<6 bp) can affect protein function. Finally, we investigated the occurrence of off-target effects in these samples by analyzing NGS data from the three most probable off-target sites per crispant, as determined by the CFD score provided by the CRISPOR tool (*Concordet and Haeussler, 2018*). In 20 out of the 30 examined off-target sites, no off-target mutations were identified (*Supplementary file 1b*). For the remaining off-target sites, the off-target rate ranged from 0,11% to 2,82%. This included basepair substitutions which are most likely sequencing errors, or true off-target indel variants, present in such low number of genomes that it is negligible in terms of biological impact in the crispants.

## Assessment of the early skeleton in crispants reveals differences in osteoblast-positive and mineralized surface areas

Crispants and sibling controls were raised until 7 or 14 dpf. At this point, the presence of immature osteoblasts in the head region was visualized using the transgenic *osx*:Kaede background. Images from 10 larvae per crispant were captured from the ventral perspective at 7 dpf and from both ventral and lateral perspectives at 14dpf, as this provides a more comprehensive view of the opercle's surface area that can be clearly visualized at that time. The osteoblast-positive areas in both the total head and the opercle were then quantified to gain insight into the formation of new skeletal tissue during early development (*Figure 1a*; *Figure 1 - figure supplement 2* and *Figure 1—figure supplement 3*).

At 7 and 14 dpf, *esr1* crispants exhibited a significant increase of the osteoblasts-positive surface area within the head (p≤0.05; p≤0.001) (*Figure 1b and c*). Additionally, *ifitm5* crispants at 14 dpf also displayed a significant increase (p≤0.01) in osteoblasts-positive surface area within the head. Finally, the increased osteoblasts-positive surface area was also observed in the opercle of both crispants at 14 dpf (*esr1* p≤0.001; *ifitm5* p≤0.01) and in the opercle of *serpinf1* (p≤0.01) and *sparc* (p≤0.05) crispants.

Subsequently, Alizarin Red S (ARS) staining was conducted on the same 7 and 14 dpf crispant zebrafish larvae in order to evaluate the degree of mineralization in the early skeletal structures. Ventral views were captured at 7 dpf, while both ventral and lateral views were obtained at 14 dpf. These lateral images at 14 dpf served a dual purpose: they facilitated measurement of the mineralized surface area of the opercle and allowed for quantification of the number of mineralized vertebrae. We quantified the mineralized surface areas of the total head, the notochord tip, and the opercle, as well as determined the number of vertebrae (*Figure 1a*; *Figure 1—figure supplement 4* and *Figure 1—figure supplement 5*).

At 7dpf, the total head exhibited a significant reduction in mineralized surface in *aldh7a1* (p≤0.05) and *sec24d* crispants (p≤0.0001), while the notochord tip displayed a significant increase in mineralized surface area in *creb3l1* (p≤0.05), *ifitm5* (p≤0.05), and *serpinf1* (p≤0.05) crispants (*Figure 1b*). At 14 dpf, a similar trend of reduced mineralized surface area of the total head was observed in *aldh7a1* and *sec24d* crispants, albeit not reaching statistical significance. In contrast, the mineral surface are of the head was significantly increased in *esr1* crispants (p≤0.05). Furthermore, the increased mineralization of the notochord tip remained significant in *ifitm5* crispants (p≤0.01), while it normalized in *creb3l1* and *sec24d* crispants. A novel increase emerged in *esr1* crispants (p≤0.05). Finally, at 14 dpf, *sec24d* crispants exhibited a significant decrease (p≤0.001) in the mineralized surface area of the opercle and the number of mineralized vertebrae, whereas *serpinf1* crispants showed a notable increase in the number of mineralized vertebrae (p≤0.01; *Figure 1c*).

## RT-qPCR expression analysis reveals differential expression of osteogenic markers *bglap* and *col1a1a* in a significant subset of crispants

To explore the expression level of different osteogenic markers throughout the set of crispants, RT-qPCR analysis was performed on whole larvae at 7 and 14 dpf. At 7 dpf, no significant differences were seen in the expression of *runx2*, an early pre-osteoblast marker, between the different crispants and corresponding controls. *Osterix,* also known as *sp7*, a *runx2* downstream gene which plays a critical role for the differentiation of pre-osteoblasts into mature osteoblasts, was only upregulated in crispants for *sparc* (p≤0.001). Osteocalcin, also known as *bglap*, which is mainly expressed in mature

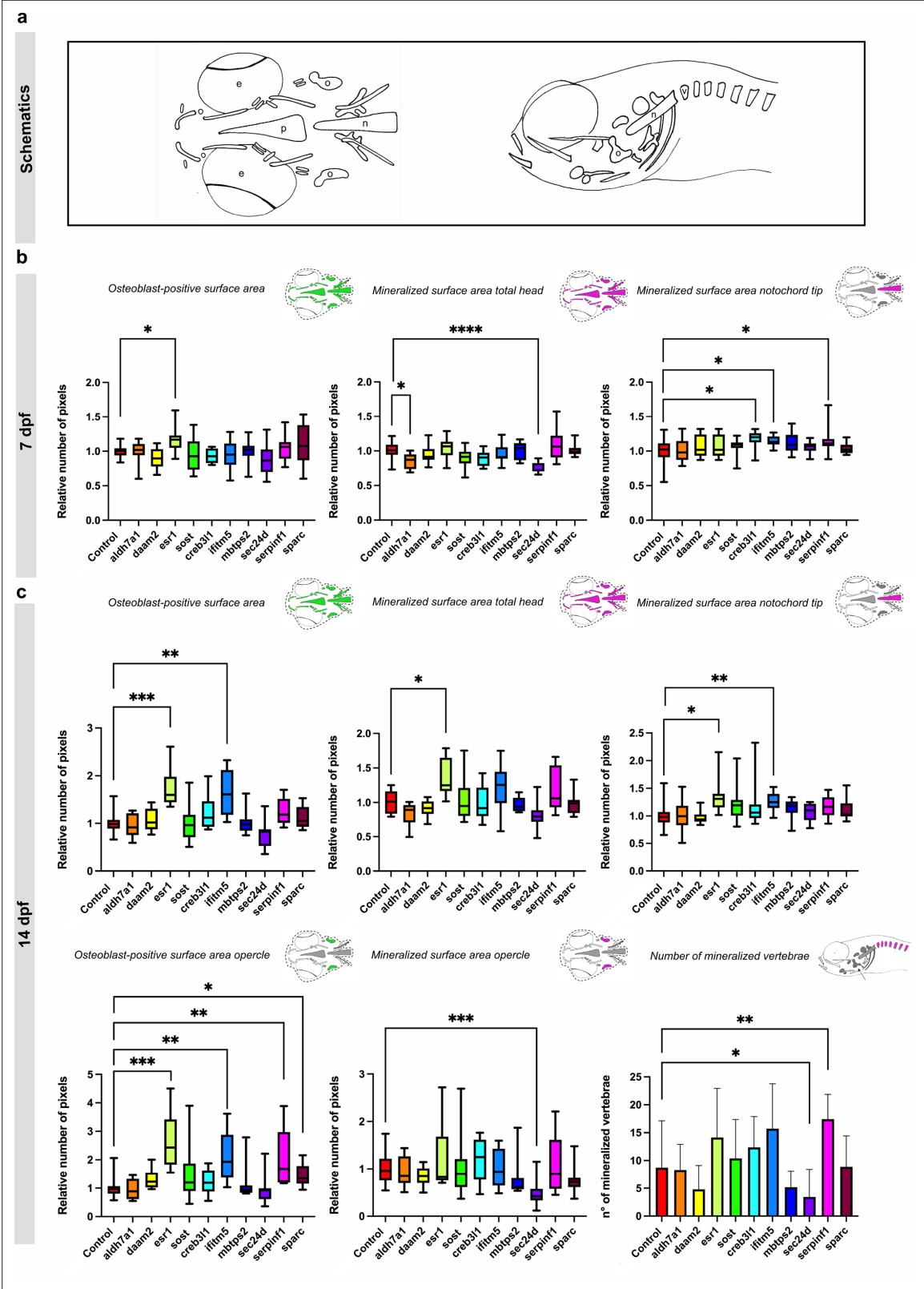

**Figure 1.** Measurements of osteoblast-positive surface area and mineralized surface area of the head skeleton in 7 and 14 dpf crispants. The first four genes are associated with the pathogenesis of osteoporosis, while the last six are linked to osteogenesis imperfecta. (**a**) Schematic overview of the ventral and lateral perspective of the head of zebrafish larvae at 7 and 14 dpf. The notochord tip (n), the opercle (o) the mineralized vertebrae (v), the eyes (e) and parasphenoid (p) are shown. (**b**) Quantification of the osteoblast-positive surface area of the total head and the mineralized surface area of

*Figure 1 continued on next page*

*Figure 1 continued*

the total head and notochord tip at 7 dpf in comparison with their respective controls. (**c**) Quantification of the osteoblast-positive surface area of the total head and the mineralized surface area of the total head, opercle and notochord tip and number of vertebrae at 14 dpf in comparison with their respective controls. For easier visualization, obtained results were normalized to the respective controls (normalization = individual value crispant (or control) / mean control) (n=10). Statistical significance is evaluated using the Mann-Whitney U -test on non-normalized data and significant differences were visualized using an asterix (*=p ≤ 0,05; **=p ≤ 0,01; ***=p ≤ 0,001; ****=p ≤ 0,0001). Error bars show the standard deviation of non-normalized data.

The online version of this article includes the following source data and figure supplement(s) for figure 1:

**Source data 1.** Raw data on osteoblast-positive surface area and mineralized surface area of the head skeleton in 7 and 14 dpf crispants.

**Figure supplement 1.** InDelphi prediction and In-frame analysis.

**Figure supplement 2.** Osteoblast-positive head area at 7 dpf.

**Figure supplement 3.** Osteoblast-positive head area at 14 dpf.

**Figure supplement 4.** Mineralization in the head area at 7 dpf.

**Figure supplement 5.** Mineralization in the head area at 14 dpf.

osteoblasts, was downregulated in almost all crispants, except in crispants for *sost* and *sparc*. Finally, *col1a1a,* encoding the α1 chain of type 1 collagen, and marking extracellular matrix synthesis by both pre- and mature osteoblasts (*Valenti et al., 2020*), was upregulated in crispants for *aldh7a1* (p≤0.001), *daam2* (p≤0.05), *esr1* (p≤0.05) and *sec24d* (p≤0.05; *Figure 2a*).

At 14 dpf, the expression of *runx2* remained unchanged in all crispants compared to control. The expression of *sp7* was significantly increased in crispants for *daam2* (p≤0.05) and *sec24d* (p≤0.05), decreased in *esr1* crispants, while expression levels were normalized in *sparc* crispants relative to 7 dpf (p≤0.05). The expression of *bglap* remained significantly downregulated in crispants for *aldh7a1* (p≤0.001), *daam2* (p≤0.01), *esr1* (p≤0.01) and *creb3l1* (p≤0.05) while it reverted to control levels in *ifitm5*, *sec24d,* and *serpinf1* crispants. An upregulation in the expression of *col1a1a* was seen in all the crispants, except in crispants for sparc and for *aldh7a1* and *esr1* where expression levels normalized compared to 7 dpf (*Figure 2b*).

## Assessment of the adult skeleton in FBD crispants reveals frequent morphological abnormalities and quantitative differences in vertebral bone parameters

Crispants were further grown into adulthood (90 dpf) and subjected to whole-mount ARS staining for mineralized bone, in order to identify skeletal abnormalities. Crispants for both *aldh7a1* and *mbtps2* displayed early lethality. Survival analysis indicated that *aldh7a1* crispants experienced increased lethality from 7 dpf onwards, with no survivors beyond 20 dpf, while *mbtps2* crispants exhibited increased lethality from 17 dpf onwards, with only a very small number of fish surviving into adulthood (90 dpf; *Figure 3*). ARS staining of the skeleton of two surviving adult *mbtps2* crispant fish revealed severe cranial malformations, including underdevelopment of the premaxilla, dentary, and articular. However, the vertebral column seemed to be only mildly affected.

Among the eight crispants that successfully matured into adulthood, none exhibited significant differences in standard length and head size (n=5 fish per crispant; *Figure 4b*). However, *esr1* crispants were observed to have notably larger eye diameters (*Figure 4b*). All these crispants demonstrated various abnormalities in the caudal part of the vertebral column such as fusions, compressions, fractures, or arch malformations, except for *daam2* crispants where no obvious abnormalities were detected (*Figure 4a and c*; *Figure 4—figure supplement 1*). In *creb3l1* crispants, the vertebral phenotype was relatively mild and primarily restricted to arch malformations.

Next, the adult skeleton of the different crispants and sibling controls was visualized by microCT scanning and the data was quantified and further analyzed using FishCuT software (*Hur et al., 2017*). For each of the vertebral centra, the length, tissue mineral density (TMD), volume, and thickness were determined and tested for statistical differences between groups using a regression-based statistical test (*Figure 4—figure supplement 2*). To quantify the extent of differences between crispants and their controls, the means of these parameters in the crispants' vertebral column, were normalized to the standard deviation of the controls, resulting in a z-score [z-score = (mean crispant – mean control) / SD control] (*Gistelinck et al., 2018*). A significant increase in centrum volume was observed in *daam2* crispants compared to the controls (p<0.05, z=2.86; *Figure 4d*). Centrum TMD was significantly

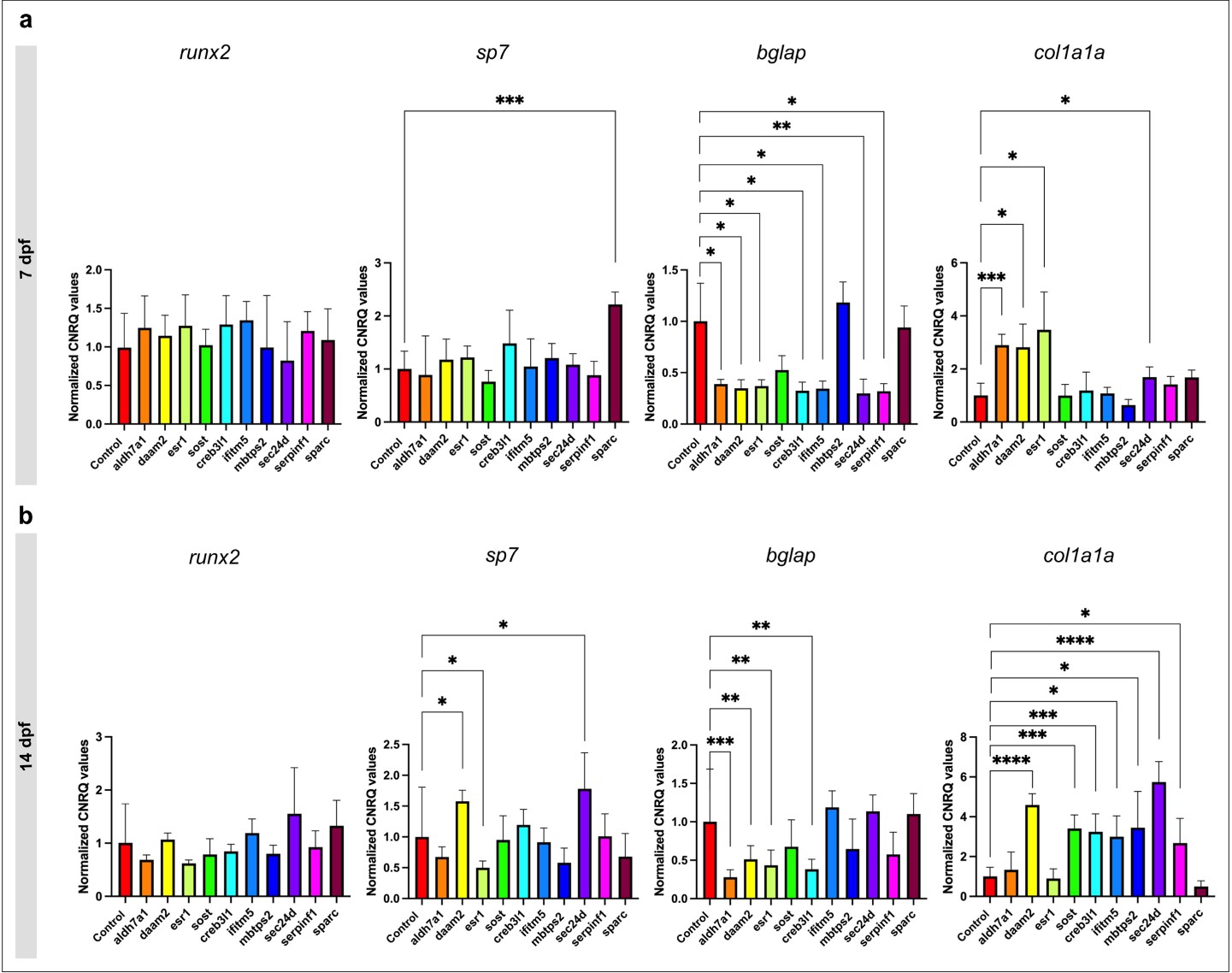

**Figure 2.** RT-qPCR expression analysis of *runx2, sp7, bglap* and *col1a1a* in crispants at 7 (**a**) and 14 dpf (**b**) and their respective controls, normalized according to the controls (normalization = individual values crispant (or control) / mean control). The first four genes are associated with the pathogenesis of osteoporosis, while the last six are linked to osteogenesis imperfecta. Statistical significance is evaluated using the Mann-Whitney U test on non-normalized data and significant differences were visualized using an asterix (*=p ≤ 0.05; **=p ≤ 0.01; ***=p ≤ 0.001; ****=p ≤ 0.0001). Error bars show the standard deviation of non-normalized data.

The online version of this article includes the following source data for figure 2:

**Source data 1.** Raw qPCR output data from the qBASE+ software (Biogazelle).

higher, and to a similar extent, in crispants for *daam2* (p<0.05; z=1.54), *esr1* (p<0.05; z=1.62), and *sost* (p<0.05; z=1.41), three genes likely involved in osteoporosis pathogenesis. Centrum thickness was increased in crispants for *esr1* (p<0.05; z=4.26), *ifitm5* (p<0.01; z=7.13), and *sec24d* (p<0.05; z=5.04) with the most pronounced increase observed in *ifitm5* crispants. Centrum length was increased only in *daam2* crispants (p<0.05; z=4.22).

## Discussion

The generation of a diverse array of null alleles through CRISPR/Cas9-induced F0 mosaic (crispant) zebrafish offers specific advantages compared to stable knockout zebrafish lines, in particular for rapid screening of phenotypes (**Naert et al., 2020**). In this study, we tested the efficacy of a semi-high

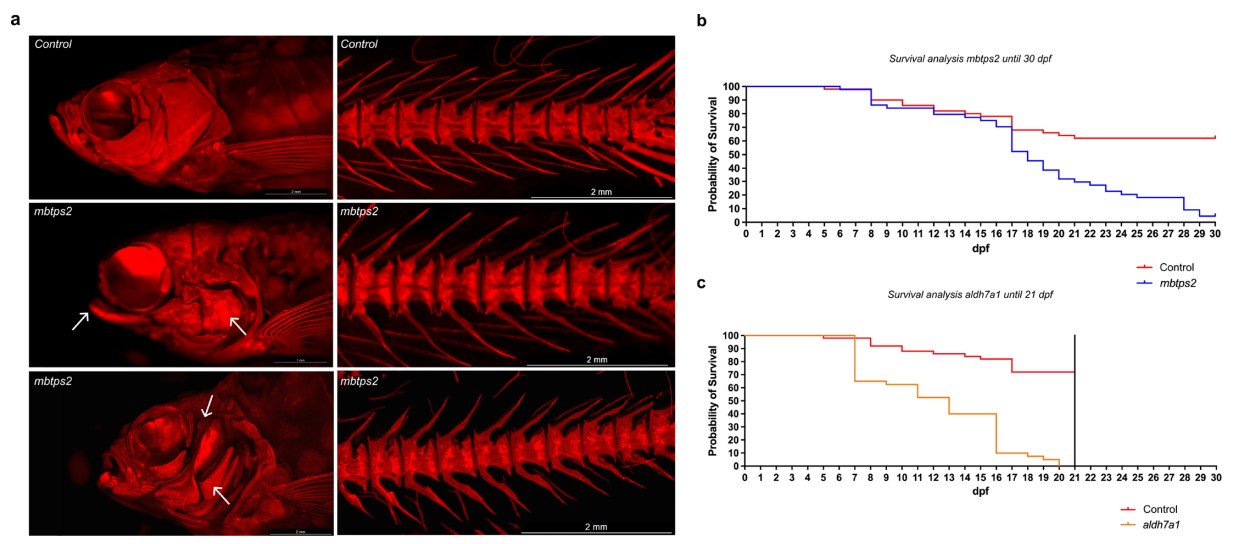

**Figure 3.** ARS images of *mbtps2* crispants and survival analysis of *aldh7a1* and *mbtps2* crispants. (**a**) ARS images of control fish and crispants for *mbtps2*, showing severe craniofacial abnormalities (arrows). (**b**) Survival curve of controls and crispants for *mbtps2*, showing a reduction in survival starting from 17 dpf. (**c**) Survival curve of controls and crispants for *aldh7a1*, showing a reduction in survival starting from 7 dpf.

The online version of this article includes the following source data for figure 3:

**Source data 1.** Raw survival curve data for *aldh7a1* and *mbtps2* crispants.

throughput zebrafish crispant analysis for rapid validation of candidate disease causing genes for skeletal disorders across a broad phenotypic spectrum. On one hand, we included genes for non-classical recessive OI, which is an early-onset monogenic skeletal syndrome with severe skeletal deformities. On the other hand, we screened a set of top candidate genes for osteoporosis, which is a rather late-onset multifactorial disease with little to no deformities. All selected genes show skeletal expression in both human and zebrafish. An overview of the selected genes with observed mutant phenotypes in human, mice and zebrafish is provided in *Supplementary file 1a*. It is important to note that this study focused on candidate genes for osteoporosis, not on the role of specific variants identified in GWAS studies. Non-coding variants for instance, which are often identified in GWAS studies, present significant challenges in terms of functional validation and interpretation. In this study we propose a crispant screening platform that combines various phenotypic and molecular assays at three key developmental and adult stages (7, 14 and 90 dpf). To obtain a general overview of the results for the selected crispants, all results were summarized in *Table 2*.

At two distinct larval stages, 7 dpf and 14 dpf, diverse measurements of the early skeleton were conducted, including an assessment of both the osteoblast-positive and the mineralized surface areas located in the head region. The osteoblast-positive area represents the quantity of osteoblasts within a given structure and is a measure of the formation of bone matrix. Although a general tendency towards increased areas could be noted across different crispants at 14 dpf, only crispants for *esr1* showed a consistent increase at both 7 and 14 dpf. Interestingly, *Brett et al., 2021* extensively investigated *esr1* mutations in the context of hormone receptor-positive breast cancer but did not evaluate the presence of osteoblasts. However, the depletion of estrogen in an osteoblastic cell line *in vitro* results in the stimulation of osteoblast differentiation (*Schiavi et al., 2021*).

The extent of mineralization of the head skeletal elements was found to be mostly not different compared to controls, although in the crispants where significant differences were detected at one of the studied time points, the mineralization of the head skeleton elements was mostly lower and of the notochord tip higher. Only crispants for *sec24d*, a gene critical for transporting extracellular matrix (ECM) proteins, exhibited reduced mineralization in different head structures both at 7 dpf and 14 dpf. These findings aligned with the observed malformations in *bulldog* zebrafish, where disrupted *sec24d* function leads to head abnormalities and growth delays in early larval stages (*Sarmah et al., 2010*). Although the measurements of osteoblast-positive and mineralized surface areas may be influenced

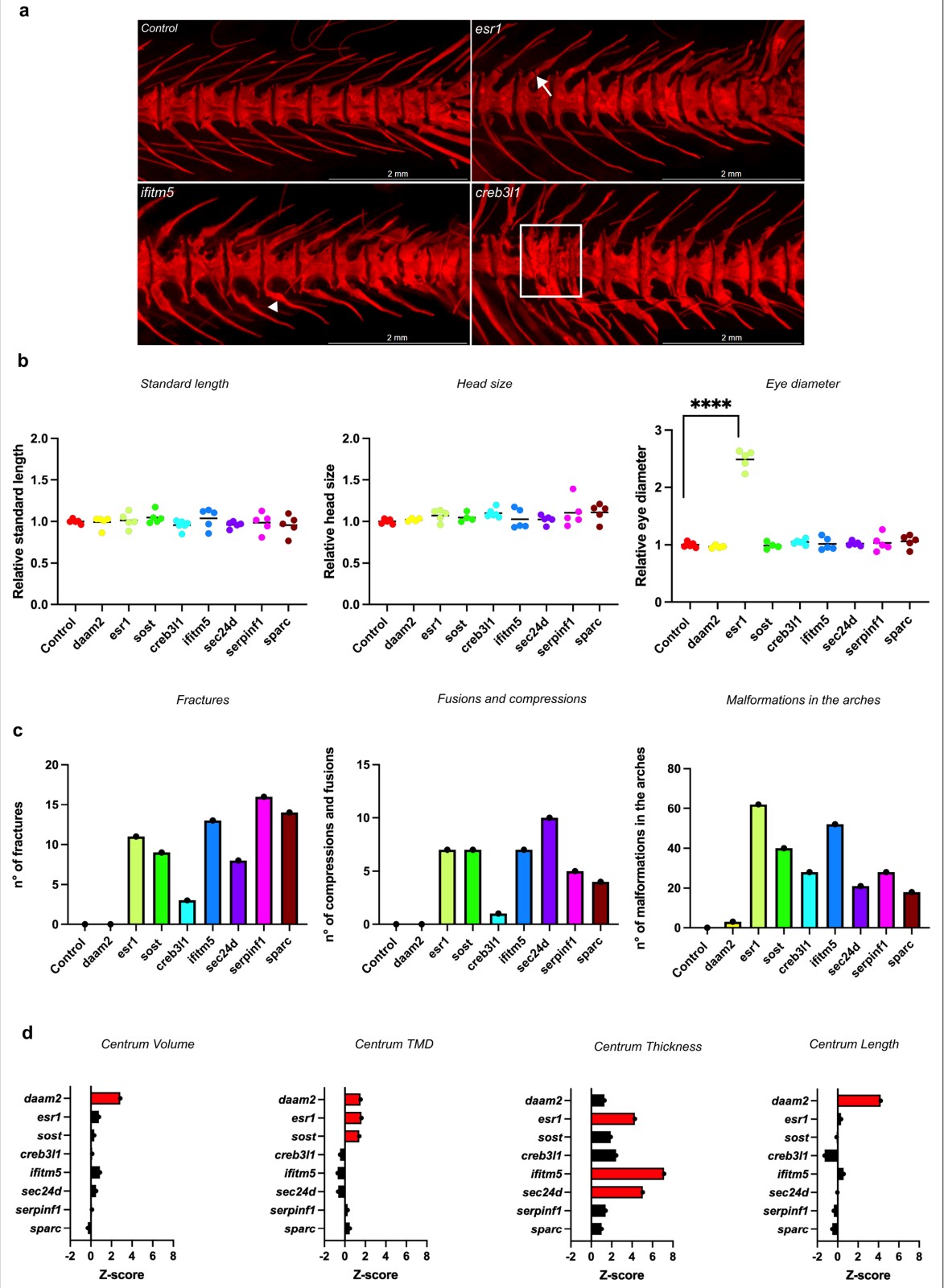

**Figure 4.** Skeletal phenotyping of adult crispants and their respective controls. The first three genes are associated with the pathogenesis of osteoporosis, while the last five are linked to osteogenesis imperfecta. (**a**) Pictures of ARS-stained vertebral column of a control and three crispants (from left to right: *esr1*, *ifitm5* and *creb3l1*), showing fractures (arrow), fusions and compressions (squared) and malformations in the arches (arrowhead). (**b**) Measurements of the standard length, the head size and the eye diameter of the crispants compared to their control. The standard length was

*Figure 4 continued on next page*

*Figure 4 continued*

measured from the snout tip to the tail base. The head size was measured from the snout tip to the supraoccipital bone and the eye diameter was measured from the lateral ethmoid to the hyomandibula. The data was normalized for easier visualization (normalization = individual value crispants (or controls) / mean control (n=5)). (**c**) Quantification of the number of fractures, fusions or compressions and malformations in the arches of crispants and the controls. For this quantification, twelve vertebrae were selected per fish and a total of 5 crispants per assay was evaluated. Statistical analysis was evaluated using the Mann-Whitney U -test on non-normalized data and significant differences were visualized using an asterix (*=p ≤ 0.05; **=p ≤ 0.01; ***=p ≤ 0.001; ****=p ≤ 0.0001). Error bars showed the standard deviation of non-normalized data. (**d**) Quantification of skeletal parameters by quantitative micro-computed tomography (μCT) analysis. In the graphical representation, the different crispants were listed. Statistically significant differences from the control values for a given crispant were depicted as red bars. Significance levels were determined through the global test analysis (*Figure 4—figure supplement 2*).

The online version of this article includes the following source data and figure supplement(s) for figure 4:

**Source data 1.** Raw data on skeletal phenotyping parameters.

**Figure supplement 1.** Mineralization in the skeleton at 90 dpf.

**Figure supplement 2.** Quantitative μCT-scanning analysis of the vertebral column of the crispants using FishCuT software.

**Figure supplement 3.** Mineralization in the skeleton at 90 dpf.

by size differences among some of the crispants, normalization to size parameters was not conducted, as variations in growth were considered integral to the phenotypic outcome.

Despite considerable variability in the measurements we conducted, a subtle trend emerged in the expression data for osteogenic markers during early stages. Specifically, the expression of *bglap*, a late osteoblast differentiation marker appeared to be consistently downregulated in near all crispants, except for crispants for *sost*, *mbtps2*, and *sparc* at 7 dpf. However, this downregulation became less pronounced at 14 dpf, affecting only crispants for *aldh7a1*, *daam2*, *esr1*, and *creb3l1*. Interestingly, most crispants for genes involved in the pathogenesis of osteoporosis exhibited this downregulation, except for *sost*, where the expression of *bglap* remained unaffected. In contrast, the expression of *col1a1a* showed an opposing trend. Initially, a modest upregulation of this marker was observed at 7 dpf, limited to just four crispants. However, this effect intensified over time, with nearly all crispants displaying upregulation at 14 dpf, except for *aldh7a1*, *esr1*, and *sparc*. The remarkable upregulation of *col1a1* expression could potentially arise from an abundance of immature osteoblasts within the cranial skeleton, a phenomenon we frequently observed across the set of crispants (*Amarasekara et al., 2021*). These osteoblasts play an active role in the synthesis of the organic bone matrix, including the production of collagen type I fibers. Interestingly, the generally diminished expression of the late osteoblastic marker *bglap* corroborates the presence of less mature osteoblasts (*Gavaia et al., 2006*). As mainly mature osteoblasts engage in the deposition of calcium and phosphate ions, essential for the formation of hydroxyapatite crystals, bone mineralization can be expected to be reduced in the crispants, which results in increased formation of non-mineralized osteoid deposits. Indeed, we observed that despite often increased bone osteoblast-positive surfaces, mineralization-positive areas in the head skeleton were frequently found to be reduced (*Salhotra et al., 2020*). The differential expression profiles of the osteogenic markers *bglap* and *col1a1a* highlight their potential as biomarkers for osteoblast differentiation and bone formation in FBD crispant screens. However, it is important to note that the genes evaluated by RT-PCR are not exclusively expressed in bone tissue. Since whole-body samples were used for expression analysis, there is a possibility that the observed changes in gene expression may be influenced by other non-skeletal cell types.

In early stages of development, we observed that crispants, although presenting with a high indel frequency, showed a relatively high variability for the phenotypic parameters we tested (mineralization and osteoblast surface areas), which often prevented reaching statistical significance. Phenotypic variability in these zebrafish larvae can be attributed to several factors, including crispant mosaicism, allele heterogeneity, environmental factors, differences in genomic background and development, maternally-deposited RNAs, and slightly variable imaging positioning. In our study, for each crispant and corresponding controls, all experiments at different ages were conducted on zebrafish from the same clutch, allowing analysis of a developmental time series in animals with a relative genetically homogeneous background. Although increasing the number of biological replicates could potentially enhance statistical power, previous studies have not confirmed this (*Winter et al., 2022*). Enhancing out-of frame (OOF) rates by simultaneously targeting a single gene with multiple crRNAs could potentially increase the penetrance of phenotypic abnormalities observed in early crispants. By directing

**Table 2.** Overview of the different measurements in crispants at 7, 14, and 90 dpf.

The first four genes are associated with the pathogenesis of osteoporosis, while the last six are linked to osteogenesis imperfecta. (-): downregulation or less, (+) upregulation or more, ns: not significant, asterix defines statistical significance (* = p ≤ 0.05; ** = p ≤ 0.01; *** = p ≤ 0.001; **** = p ≤ 0.0001).

| | aldh7a1 | daam 2 | esr1 | sost | creb3l1 | ifitm 5 | mbtps2 | sec24d | serpinf1 | sparc |
|---|---|---|---|---|---|---|---|---|---|---|
| **Osteoblast-positive surface area total head** | | | | | | | | | | |
| 7 dpf | ns | ns | (+)* | ns | ns | ns | ns | ns | ns | ns |
| 14 dpf | ns | ns | (+)*** | ns | ns | (+)** | ns | ns | ns | ns |
| **Osteoblast-positive surface area operculum 14 dpf** | ns | ns | (+)*** | ns | ns | (+)** | ns | ns | (+)** | (+)* |
| **Mineralized surface area total head** | | | | | | | | | | |
| 7 dpf | (–)* | ns | ns | ns | ns | ns | ns | (–)**** | ns | ns |
| 14 dpf | (–)* | ns | ns | ns | ns | ns | ns | ns | ns | ns |
| **Mineralized surface area notochord tip** | | | | | | | | | | |
| 7 dpf | ns | ns | ns | ns | (+)* | (+)* | ns | (+)* | ns | ns |
| 14 dpf | ns | ns | (+)** | ns | ns | (+)* | ns | ns | ns | ns |
| **Mineralized surface area operculum 14 dpf** | ns | ns | ns | ns | ns | ns | ns | (–)** | ns | ns |
| **Number of mineralized vertebrae 14 dpf** | ns | ns | ns | ns | ns | ns | ns | (–)** | ns | ns |
| ***runx2* expression** | | | | | | | | | | |
| 7 dpf | ns | ns | ns | ns | ns | ns | ns | ns | ns | ns |
| 14 dpf | ns | ns | ns | ns | ns | ns | ns | ns | ns | ns |
| ***sp7* expression** | | | | | | | | | | |
| 7 dpf | ns | ns | ns | ns | ns | ns | ns | ns | ns | (+)*** |
| 14 dpf | ns | (+)* | (–)* | ns | ns | ns | ns | (+)* | ns | ns |
| ***bglap* expression** | | | | | | | | | | |
| 7 dpf | (–)* | (–)* | (–)* | ns | (–)* | (–)* | ns | (–)** | (–)* | ns |
| 14 dpf | (–)*** | (–)** | (–)** | ns | (–)** | ns | ns | ns | ns | ns |
| ***col1a1a* expression** | | | | | | | | | | |
| 7 dpf | (+)*** | (+)* | (+)* | ns | ns | ns | ns | (+)* | ns | ns |
| 14 dpf | ns | (+)**** | ns | (+)*** | (+)*** | (+)* | ns | (+)**** | (+)* | ns |
| **Standard length 90 dpf** | | ns | ns | ns | ns | ns | | ns | ns | ns |
| **Head size 90 dpf** | | ns | ns | ns | ns | ns | | ns | ns | ns |
| **Diameter eye 90 dpf** | | ns | ns | ns | ns | ns | | ns | ns | ns |
| **Centrum volume 90 dpf** | | (+)* | ns | ns | ns | ns | | ns | ns | ns |
| **Centrum TMD 90 dpf** | | (+)* | (+)* | (+)* | ns | ns | | ns | ns | ns |
| **Centrum thickness 90 dpf** | | ns | (+)* | ns | ns | (+)* | | (+)* | ns | ns |
| **Centrum Length 90 dpf** | | (+)* | ns | ns | ns | ns | | ns | ns | ns |

crRNAs to multiple loci within one gene, the likelihood of introducing frameshift mutations increases significantly. This is suggested to lead to a higher rate of biallelic knockout and reduce the chances of retaining the functional protein (*Kroll et al., 2021*). However, this also elevates the likelihood of off-target effects, increases double-strand breaks and toxicity, while simultaneously increasing costs (*Wu et al., 2018*). Furthermore, also in-frame indels affecting two or more amino acids, detected in a substantial proportion of NGS reads in our study, are likely to have a detrimental effect on protein

function (*Emond et al., 2020*; *Savino et al., 2022*; *Miton and Tokuriki, 2023*). Therefore, and because we used highly efficient two-part guide RNAs Alt-R CRISPR-Cas9 system, we opted for single crRNA injections. While there remains a possibility of false negatives, the overall indel efficiency, as indicated by our NGS analysis, is high (>90%), thereby reducing the likelihood of having crispants with very low indel efficiency.

In adult crispants, the skeletal phenotype was generally more penetrant. All crispants showed malformed arches, a majority displayed vertebral fractures and fusions and some crispants exhibited distinct quantitative variations in vertebral body measurements. Additionally, these skeletal malformations were consistently observed in a second clutch of crispants (*Figure 4—figure supplement 3*), underscoring the reproducibility of these phenotypic features across independent clutches. This confirmed the role of the selected genes in skeletal development and homeostasis and their involvement in skeletal disease and established the crispant approach as a valid approach for rapidly providing *in vivo* gene function data to support candidate gene identification.

Osteoporosis is typically linked with symptoms that appear late in life. However, our results challenged this notion, by showing that in crispants for osteoporosis genes, skeletal abnormalities can manifest earlier and are not solely reflected in changes to bone mineral density (BMD). For instance, we found that *aldh7a1* and *esr1* crispants displayed significant skeletal anomalies already at larval stages, with *aldh7a1* crispants showing early lethality. At adult stages, *esr1* and *sost* crispants demonstrated various morphological abnormalities of the vertebral column such as fusions, compressions, and arch malformations. The more pronounced and earlier phenotypes in these zebrafish crispants are most likely attributed to the quasi knock-out state of the studied genes, while more common less impactful variants in the same genes result in typical late-onset osteoporosis (*Laine et al., 2013*). This phenomenon is also observed in knock-out mouse models for these genes (*Melville et al., 2014*; *Coughlin et al., 2019*).

Interestingly, all adult crispants for osteoporosis genes demonstrated a substantial increase in centrum Tissue Mineral Density (TMD). TMD is quantitative measure of the actual mineral volume within bone tissue, computed as bone mineral content over the bone volume (mgHA/cm3) and obtained through mCT or pQCT. In human patients, reduced bone mineral density (aBMD), a measure for the bone mineral content in a projected bone area and assessed using dual energy X-ray absorptiometry (DXA), is frequently linked to osteoporosis. However, also TMD, or vBMD, can be evaluated in the trabecular and cortical bone. Knock-out mice lacking Sost showed an increase in TMD within the cortical bone, which is comparable to the bone present in the zebrafish vertebral column (*Li et al., 2008*). As for the other genes, for *daam2*, mutant mice with a hypomorphic *Daam2* allele presented with reduced bone strength and increased cortical bone porosity (*Morris et al., 2019*). *ERα* knockout mice presented with a decrease in cortical bone mineral density and increase in trabecular bone mineral density (*Melville et al., 2014*). Further studies are needed to better understand how high TMD values observed in these crispants relate to phenotypic changes observed in these mouse models. Finally, it is worthwhile to note that, osteoporosis typically correlates with low vBMD values. While neither the causal variant nor the biological direction of its effects on the target gene are known for BMD-associated loci harboring these genes, the increased TMD following gene knockdown in our models may help to assign the directionality of their effects, and puts forth the possibility that variants at these loci alter the expression of genes that converge on a common biological process in zebrafish, when disrupted results in the high TMD.

Conversely, crispants for genes involved in recessive forms of OI, an early-onset severe skeletal disorder, can intuitively be expected to exhibit pronounced skeletal abnormalities already in larval stages. Interestingly, during these early stages, the majority of the crispants do not exhibit consistent skeletal abnormalities. Although this could be partially explained by the above-discussed variability of the phenotypic outcomes we measured, also human OI subjects with these severe conditions, often experience disease onset during infancy only, with progressive worsening of the skeletal phenotype (*Selina et al., 2023*).

The phenomenon of genetic compensation could play a role in explaining why early phenotypes manifest differently in human patients compared to zebrafish. Genetic compensation ensures an organism to maintain its genetic robustness, even when genetic variation, possibly induced by mutations, occurs. This compensation mechanism can be triggered by the upregulation of compensating genes, often mediated through the regulation of nonsense-mediated decay (NMD) in combination

with the epigenetic machinery in zebrafish (*El-Brolosy and Stainier, 2017*; *Jakutis and Stainier, 2021*). Crispants, in contrast to morphants, also undergo NMD although in a mosaic manner. In larval crispants, compensatory mechanisms may effectively mask the impact of gene loss. However, in adult crispants, the cumulative effect of this loss becomes more pronounced, leading to a later onset phenotype. Furthermore, specific tissues or cell types within the organism may activate compensatory pathways more efficiently, thereby influencing the overall manifestation of the phenotype. When a gene is disrupted in humans, the consequences can be more severe due to the complexity of human development and biology. Such disruptions may result in developmental abnormalities, leading to an early onset of phenotypic traits (*Rouf et al., 2023*).

To conclude, our study proposes a multifaceted gene-based approach to investigate zebrafish crispants associated with heritable FBDs, employing various techniques, and assessing different skeletal phenotypes and molecular profiles at distinct developmental stages. While this work represents a pioneering effort in establishing a screening platform for skeletal diseases, it offers opportunities for future improvement and refinement. For instance, analyzing different skeletal structures and cell types within various transgenic lines, increasing sample size, and incorporating a broader range of techniques, such as phenotypic scoring based on the type of malformations seen in the vertebral column or alcian blue staining for cartilage tissue evaluation, could enhance the comprehensiveness of the screening process and the skeletal phenotype. Moreover, to explore the underlying mechanisms contributing to disease phenotypes, it is essential to establish stable knockout mutants derived from the crispants. Nonetheless, our initial set-up demonstrates the feasibility of this approach to identify genes potentially implicated in the pathogenesis of skeletal disorders, as all studied crispants exhibit skeletal abnormalities in adult stages. These findings support the prospect of crispant screening as a promising tool for rapid functional assessment of candidate genes for skeletal disorders, providing valuable insights into the roles of these genes in skeletal biology.

## Materials and methods

**Key resources table**

| Reagent type (species) or resource | Designation | Source or reference | Identifiers | Additional information |
|---|---|---|---|---|
| Genetic reagent (*Danio rerio*) | osx:Kaede transgenic line | Zebrafish Facility Ghent (ZFG) Core Facility | NA | NA |
| Commercial assay or kit | RNeasy Mini Kit 250 reagentia | Qiagen | 74106 | NA |
| Commercial assay or kit | iScript cDNA Synthesis Kit | Bio-rad | 1708891 | NA |
| Commercial assay or kit | SsoAdvanced Universal SYBR Green Supermix | Bio-rad | 172–5274 | NA |
| Other | Alizarin red S staining | Merck life sience bv | A5533-25G | NA |

### Zebrafish maintenance and breeding

The zebrafish were incubated at 28 °C until 5 days post-fertilization (dpf). Then, they were reared in a rotifer/zebrafish polyculture system from 5dpf to 9dpf following the method of *Lawrence et al., 2016*. After 10dpf, they were transferred to 3.5 liter tanks in the Core Zebrafish Facility Ghent (ZFG) and housed in semi-closed recirculating systems (Tecniplast). The water parameters were maintained at 27 °C, pH 7.5, conductivity 550 µS, and a 14/10 light/dark cycle. The zebrafish were fed with Zebrafeed (Sparos) and Gemma Micro dry food (Inve) in the morning and Micro Artemia (Ocean Nutrition) in the afternoon. The *osx*:Kaede transgenic line was used for all experiments. Breeding and embryo collection were done according to standard protocols (*Westerfield, 2000*). All animal studies complied with EU Directive 2010/63/EU for animals, permit no. ECD 23/27 (Ghent University). Pain, distress, and discomfort were minimized as much as possible.

### Generation of crispants through CRISPR/Cas9 genome editing
#### Guide RNA design
For the generation of crispants we largely followed the methods described in *Bek et al., 2020*. In summary, two-part guide RNAs (gRNAs) were used which are functional gRNA duplexes made of a

universal 67 mer trans-activating CRISPR-RNA (tracrRNA) and a target-specific CRISPR-RNA (crRNA) (Alt-R CRISPR-Cas9 System, IDT). A non-specific crRNA (GCAGGCAAAGAATCCCTGCC), referred to as the 'scrambled' sequence, was used to generate the controls (*Wu et al., 2018*).

The crRNAs corresponding to the genes of interest were designed with the web-based tool Benchling (https://benchling.com), using GRCz11 as a reference genome. With this tool, 20 bp crRNAs target sequences, that are followed by a 'NGG' PAM sequence, were listed. Further selection was based on a few specific criteria to minimize the risk of producing (partially) active truncated protein products and potential off-target effects. Specifically, crRNA target sequences located in exon 1, in the last exon, within the last 55 bp of the second-to-last exon or in exons consisting of (n) x 3 bp were excluded. Additionally, crRNAs with the highest possible scores for both on-target (*Doench et al., 2016*) and off-target (*Hsu et al., 2013*) were selected.

For the selected crRNA sequences, the Indelphi tool (https://indelphi.giffordlab.mit.edu) employing the mESC training set, was used to predict the DNA repair outcomes (mixture of indels) resulting from non-homologous end-joining (NHEJ) and microhomology-mediated end-joining (MMEJ) induced by CRISPR/Cas9 cleavage. The crRNAs that were predicted to result in high percentages of out-of-frame indels and a limited number of possible genotypes (high precision) were prioritized.

Eventually, 1 crRNA per gene was ordered from IDT (https://www.idtdna.com): *aldh7a1* (AATT GTTCGACAGATTGGAG), *daam2* (GATCTATTGCAGTAAGAAGA), *esr1* (GCTCACGACAGAAACA CAGC), *sost* (CTGCTCACTCCCGCACCCAG), *creb3l1* (ACACAGTTACTCTCTCAGCG), *ifitm5* ( AGCGCAGGAATGCTCAGACA), *mbtps2* (TTCCATATCAAGTGGCACAC), *sec24d* (GCCTATGG ATCTCCAACACA), *serpinf1* (CAGGTTGTAGCCAAAATCAG), *sparc* (AGAGGAGCCAGCTGTT GAAG).

To predict the potential off-target effects of the selected crRNAs, Crispor (http://crispor.tefor.net) was used. The cutting frequently determination (CFD) off-target score was used to predict the probability of off-target cleavage. The three targets with the highest CFD scores were selected and PCR-amplified (*Supplementary file 1b*). Next-generation sequencing using the Miseq platform was used to assess the frequency of the off-target effects (*Concordet and Haeussler, 2018*).

## Injection

To prepare gRNA duplexes for each assay, we mixed 200 µM crRNA and 200 µM tracrRNA, heated the mixture at 95 °C for 5 min, and then cooled it to room temperature. Next, we micro-injected one-cell-stage Tg(*osx*:Kaede) zebrafish embryos. The injection mix consisted of a 1.4 nL drop containing RNase-free water, KCl (2000 mM) and ribonucleoproteins (RNPs), consisting of 800 pg of both gRNA duplex and Cas9 (1000 ng/µl) (ToolGen, Seoul, South Korea; wild-type nuclease with nuclear localization signal) protein. For each crispant and each corresponding control, a total of 200 eggs, originating from the same clutch, were injected.

## Next-generation sequencing analysis

We extracted DNA from 10 pooled embryos at one day after fertilization by adding 100 µl NaOH and incubating the samples at 95 °C for 20 min. We then added 10 µL Tris-HCl. We performed PCR amplification with the primers listed in *Supplementary file 1c*, using either 'FORD58' or 'FORD60' assays depending on the primer. The thermocycling conditions for the 'FORD58' assay were: 95 °C for 3 min, followed by 40 cycles of 95 °C for 15 s, 58 °C for 10 s, and 72 °C for 15 s, and a final extension at 72 °C for 10 min. The thermocycling conditions for the 'FORD60' assay were: 95 °C for 3 min, followed by 40 cycles of 95 °C for 15 s, 60 °C for 10 s, and 72 °C for 15 s, and a final extension at 72 °C for 10 min. We prepared the PCR amplicons for sequencing using the Nextera XT DNA Library Preparation Kit (Illumina, San Diego, CA, USA) and sequenced them on a MiSeq instrument (Illumina) using 2X250 bp cycles (*Bek et al., 2020*). We analyzed the NGS data with CRISPResso2 and determined the mean fraction of reads containing in-frame and out-of-frame indels over different samples, by exporting the allele frequency table, counting the number of reads with both in- and out-of-frame indels or out-of-frame indels only and dividing these numbers by the total number of mapped reads (*Fiume et al., 2019*). For both the indel and out-of-frame efficiency, the reads with sequencing errors and natural variants were excluded.

## RNA extraction and RT-qPCR

The RNA extraction and qPCR protocol described by *Bek et al., 2020* was followed, with minor modifications; for each crispant, RNA was extracted from 4 or 5 pools of 10 embryos at 7 dpf and 14 dpf using Trizol (Life Technologies Europe) and purified using the RNeasy Mini Kit (Qiagen) with on-column DNase I treatment (Qiagen). RNA concentrations were measured with the Little Lunatic (Unchained Labs) and cDNA was synthesized using the iScript cDNA synthesis kit (Bio-rad). Primers for skeletal marker genes were designed using NCBI Primer-BLAST (*Supplementary file 1d*) and primers for the housekeeping genes (*elfa* and *bactin2*) were obtained from literature (*McCurley and Callard, 2008*).

The qPCR reaction mixture was prepared with 2.5 µL of SsoAdvanced Universal SYBR Green Supermix (Bio-rad), 5 ng of cDNA, and 250 nM of each primer. The qPCR reaction was run on a Light-Cycler 480 instrument (Roche) in a white 384-well plate with the following thermocycling conditions: initial denaturation step at 95 °C for 2 min, followed by 44 cycles of denaturation at 95 °C for 5 s, annealing at 60 °C for 30 s, and extension at 72 °C for 1 s. Subsequently, a melting curve analysis was performed with a denaturation step at 95 °C for 5 s, followed by annealing at 60 °C for 1 min and gradual heating to 95 °C at a ramp-rate of 0.11 °C/s. Finally, the product was cooled to 37 °C for 3 min (*Vanhauwaert et al., 2014*). The data was analyzed using the qBase +software (Biogazelle).

## Osteoblast imaging in transgenic *osx:*Kaede larvae

Ten zebrafish larvae (at 7 dpf and 14 dpf) with the transgenic *osx:*Kaede background were anesthetized using 1 x Tricaine. Imaging was performed using the Leica microscope M165FC and mounted camera (DFC450C) operated by LAS V4.3 software (Leica Microsystems, Wetzlar, Germany). Images of 7 dpf larvae were taken from the ventral side. Images of 14 dpf larvae were taken from both ventral and lateral perspectives to provide a more comprehensive view of the opercle's surface area.

Fiji software (version 2.14.0), powered by ImageJ, was employed for all measurements (*Schindelin et al., 2012*). Osteoblast-positive area of both the total head and opercle was quantified using thresholding, followed by measurement of the positive area. All measurements were performed in triplicate on blinded pictures. The presence of osteoblasts in the opercle was measured by hand using the polygon selection tool. Mann-Whitney U test was applied in GraphPad Prism (v.9.4.0), with p-values considered significant when ≤0.05.

## Alizarin red staining and imaging of zebrafish larvae

The ten zebrafish larvae, that were previously used for osteoblast imaging at 7 dpf and 14 dpf, were euthanized using 25 x Tricaine. They were transferred to a cell strainer and washed with dH2O several times to remove the Tricaine. The strainers were placed in a 6-well plate and incubated in 4% PFA overnight at 4 °C. The plates were sealed with parafilm to prevent evaporation. The larvae were then washed with dH2O several times to remove the PFA and bleached for 30 min at room temperature using a solution containing 3% $H_2O_2$ and 2% KOH to remove pigmentation. Next, they were stained overnight at 4 °C using Alizarin Red S (0.005% in 1% KOH/2% Triton). The samples were washed with dH2O several times and transferred to increasing concentrations of glycerol (30-70–100% for 5 min each) for storage (4 °C) and imaging. Zebrafish larvae stained with Alizarin red were then imaged using a Leica M165FC microscope and DFC450C camera with the LAS V4.3 software (Leica Microsystems). The ventral side of the larvae was captured at 7 dpf. Both ventral and lateral perspectives were captured at 14 dpf to provide a more comprehensive view of the opercle's surface area and the number of mineralized vertebrae.

Fiji software (version 2.14.0), powered by ImageJ, was employed for all measurements (*Schindelin et al., 2012*). The red stain in the head elements was assessed using thresholding, followed by quantification of the positive area. These measurements were conducted in triplicate on blinded pictures. The stained notochord tip and opercle was measured by hand using the polygon selection tool. The statistical analysis employed the Mann-Whitney U test in GraphPad Prism (v.9.4.0), with p-values considered significant when ≤0,05.

## Micro-computed tomography (MicroCT) analysis in adult zebrafish

Adult zebrafish (3 months post fertilization) were euthanized using a 25 x tricaine solution (4000 mg tricaine powder (Sigma), 979 ml distilled water, and 1 M Tris, adjusted to pH 7). Brightfield images of the zebrafish were taken using a Leica M165FC microscope and DFC450C camera with the LAS V4.3

software (Leica Microsystems). The standard lengths of the zebrafish were measured using the Fiji (ImageJ) software (ImageJ 1.52i) (*Schindelin et al., 2012*).

The fish were fixed for approximately one week in a fixative solution (4% paraformaldehyde, 5% triton X-100, and 1% potassium hydroxide). The solution was changed when it turned yellow. A maximum of five fish per 100 ml of fixative solution was used. After fixation, the fish were transferred to a 70% ethanol solution for storage until scanning.

Before scanning, three fish were transferred to a 15 ml Eppendorf tube and immobilized using paper towel soaked in 70% ethanol. The Eppendorf tubes were scanned at the custom-built HECTOR scanner of the Centre for X-ray Tomography (UGCT, https://www.ugent.be/we/ugct/en). A hardware filter of 0.5 mm Al was used to eliminate beam hardening artefacts, with the tube voltage at 90 kVp and target power of 10 W. During a 360° rotation, 1201 radiographic images were acquired at 1 s acquisition time per image. The raw data was reconstructed using the Octopus Reconstruction software tool, resulting in a 3D volume of approximately 2800x1000 × 1000 voxels. Using geometrical magnification, a voxel size of $20^3$ µm³ was obtained (*Masschaele et al., 2013*).

The 3 fish in one tube were virtually separated from the scan, aligned and exported as 3 individual single tiff volume using VGStudioMAX (VolumeGraphics, Heidelberg, Germany). The generated volumes were analyzed using the FishCut software in Matlab described in *Hur et al., 2017*. The resulting plots were generated using R statistics and were color-coded to highlight significant differences. Significant differences were represented with a lighter color scheme.

## Alizarin red staining and imaging of adult zebrafish

The fish used for microCT were reused for Alizarin Red staining for which the protocol has been described in *Bek et al., 2020*. In short, the fish were transferred to cassettes and the concentration of EtOH was gradually reduced from 70% to distilled water (70%–50%–30%-dH2O each 30 min). Then, the samples were enhanced, immersed into the bone-staining medium, stained with bone-staining solution and destained after removal of the scales with destaining solution. Finally, the samples were gradually (30-70–100%) transferred to 100% glycerol for storage (4 °C) and imaging.

Adult zebrafish were imaged using a Leica M165FC microscope and DFC450C camera with the LAS V4.3 software (Leica Microsystems). The DSR DsRed Dichroic Mirror filter was used to visualize the (red) fluorescent signal. Images were obtained from a lateral view to get a clear picture of the whole fish. Higher magnification was used to get detailed images of the vertebral column and the head. Images of the head were used to measure the head size, using the Fiji (ImageJ) software (*Schindelin et al., 2012*).

## Acknowledgements

SD acknowledges funding from the fonds wetenschappelijk onderzoek (FWO) FWO.OPR.2020.0023.01. We would like to thank the Zebrafish Facility Ghent (ZFG) Core Facility at Ghent University.

## Additional information

### Funding

| Funder | Grant reference number | Author |
| --- | --- | --- |
| Fonds Wetenschappelijk Onderzoek | FWO.OPR.2020.0023.01 | Sophie Debaenst |

The funders had no role in study design, data collection and interpretation, or the decision to submit the work for publication.

### Author contributions

Sophie Debaenst, Conceptualization, Data curation, Formal analysis, Validation, Investigation, Visualization, Methodology, Writing - original draft, Project administration, Writing – review and editing; Tamara Jarayseh, Jan Willem Bek, Conceptualization, Writing – review and editing; Hanna De Saffel, Conceptualization, Data curation, Methodology; Matthieu Boone, Software, Methodology, Writing

– review and editing; Ivan Josipovic, Data curation, Software, Methodology; Pierre Kibleur, Software, Writing – review and editing; Ronald Y Kwon, Conceptualization, Software, Formal analysis, Writing – review and editing; Paul J Coucke, Conceptualization, Resources, Supervision, Funding acquisition, Methodology, Project administration, Writing – review and editing; Andy Willaert, Conceptualization, Supervision, Funding acquisition, Writing – review and editing

## Author ORCIDs
Sophie Debaenst ⓘ https://orcid.org/0000-0001-7598-919X
Ronald Y Kwon ⓘ https://orcid.org/0000-0001-9760-3761
Andy Willaert ⓘ https://orcid.org/0000-0002-9543-1932

## Ethics
All animal experiments were performed according to EU Directive 2010/63/EU, permit no. ECD 23/27 (Ghent University). Pain, distress, and discomfort were minimized as much as possible.

Reviewer #1 (Public review): https://doi.org/10.7554/eLife.100060.3.sa1
Reviewer #2 (Public review): https://doi.org/10.7554/eLife.100060.3.sa2
Reviewer #3 (Public review): https://doi.org/10.7554/eLife.100060.3.sa3
Author response https://doi.org/10.7554/eLife.100060.3.sa4

## Additional files

### Supplementary files
Supplementary file 1. Supplementary material: detailed information on human and mouse mutations, off-target effects, and sequences for genotyping and qPCR analysis. (a) Overview of selected genes for crispant analysis, with reported mutations and/or polymorphisms associated with skeletal and non-skeletal phenotypes in human, mice and zebrafish. The conservation between human and zebrafish is reported in the last column. The first four genes are associated with the pathogenesis of osteoporosis, while the last six are linked to osteogenesis imperfecta. (b) Assessment of off-target effects in crispants. Genes that are possibly targeted by the selected crRNA for each of the crispants are called 'off-target genes' in this table (mm=number of mismatches). The top 3 ranked off-target effects, selected based on a high CFD (cutting frequency determination) score, are listed, together with their chromosomal position, forward and reverse primer for amplification, the CFD (cutting frequency determination) score and the off-target percentage, based on NGS analysis of a pool of DNA of 1-day old crispants (n=10). (c) Crispant genotyping. Crispant genes with crRNA sequence, forward and reverse primers and assay specifications for Next-generation sequencing (NGS) are listed in this table. The first four genes are associated with the pathogenesis of osteoporosis, while the last six are linked to osteogenesis imperfecta. Primers are designed using Primer3 (https://primer3.ut.ee). (d) qPCR primers. Skeletal marker genes and reference genes with forward and reverse primers are listed in this table. Primers are designed using NCBI PrimerBlast (https://www.ncbi.nlm.nih.gov/tools/primer-blast).

MDAR checklist

### Data availability
*Figure 1—source data 1*, *Figure 2—source data 1*, *Figure 3—source data 1* and *Figure 4—source data 1* contain the numerical data used to generate the figures.

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
