## [Editor Report · eLife Assessment]

The paper presents a streamlined new approach for functional validation of genes known to underlie fragile bone disorders in a relatively high throughput, using CRISPR-mediated knockouts and a number of phenotypic assessments in zebrafish. **Convincing** data demonstrate the feasibility and validity of this approach, which presents an **important** tool for rapid functional validation of candidate gene(s) associated with heritable bone diseases identified from genetic studies.

---

## [Referee Report · Reviewer #1 (Public review)]

Summary:

In this work, a screening platform is presented for rapid and cost-effective screening of candidate genes involved in Fragile Bone Disorders. The authors validate the approach of using crispants, generating FO mosaic mutants, to evaluate the function of specific target genes in this particular condition. The design of the guide RNAs is convincingly described, while the effectiveness of the method is evaluated to 60% to 92% of the respective target genes being presumably inactivated. Thus, injected F0 larvae can be directly used to investigate the consequences of this inactivation.

Skeletal formation is then evaluated at 7dpf and 14dpf, first using a transgenic reporter line revealing fluorescent osteoblasts, second using alizarin-red staining of mineralized structures. In general, it appears that the osteoblast-positive areas are more often affected in the crispants compared to the mineralized areas, an observation that appears to correlate with the observed reduced expression of bglap, a marker for mature osteoblasts, and the increased expression of col1a1a in more immature osteoblasts.

Finally, the injected fish (except two lines that revealed a high mortality) are also analyzed at 90dpf, using alizarin red staining and micro-CT analysis, revealing an increased incidence of skeletal deformities in the vertebral arches, fractures, as well as vertebral fusions and compressions for all crispants except those for daam2. Finally, the Tissue Mineral Density (TMD) as determined by micro-CT is proposed as an important marker for investigating genes involved in osteoporosis.

Taken together, this manuscript is well presented, the data are clear and well analyzed, and the methods well described. It makes a compelling case for using the crispant technology to screen the function of candidate genes in a specific condition, as shown here for bone disorders.

Strengths:

Strengths are the clever combination of existing technologies from different fields to build a screening platform. All the required methods are comprehensively described.

Weaknesses:

One may have wished to bring one or two of the crispants to the stage of bona fide mutants, to confirm the results of the screening, however, this is done for some of the tested genes as laid out in the discussion.

Comments on latest version:

All my issues were resolved.

---

## [Referee Report · Reviewer #2 (Public review)]

Summary:

More and more genes and genetic loci are being linked to bone fragility disorders like osteoporosis and osteogenesis imperfecta through GWAS and clinical sequencing. In this study, the authors seek to develop a pipeline for validating these new candidate genes using crispant screening in zebrafish. Candidates were selected based on GWAS bone density evidence (4 genes) or linkage to OI cases plus some aspect of bone biology (6 genes). NGS was performed on embryos injected with different gRNAs/Cas9 to confirm high mutagenic efficacy, and off-target cutting was verified to be low. Bone growth, mineralization, density, and gene expression levels were carefully measured and compared across crispants using a battery of assays at three different stages.

Strengths:

(1) The pipeline would be straightforward to replicate in other labs, and the study could thus make a real contribution towards resolving the major bottleneck of candidate gene validation.

(2) The study is clearly written and extensively quantified.

(3) The discussion attempts to place the phenotypes of different crispant lines into the context of what is already known about each gene's function.

(4) There is added value in seeing the results for the different crispant lines side by side for each assay.

(5) Caveats to the interpretability of crispant data and limitations of their gene-focused analyses and RT-PCR assays are discussed.

Weaknesses:

(1) The study uses only well-established methods and is strategy-driven rather question/hypothesis-driven. This is in line with the researchers' primary goal of developing a workflow for rapid in vivo functional screening of candidate genes. However, this means that less attention is paid to what the results obtained for a given gene may mean regarding potential disease mechanisms, and how contradictions with prior reports of mouse or fish null mutant phenotypes might be explained.

(2) Normalization to body size was not performed. Measurements of surface area covered by osteoblasts or mineralized bone (Fig. 1) are typically normalized to body size - especially in larvae and juvenile fish - to rule out secondary changes due to delayed or accelerated overall growth. This was not done here; the authors argue that "variations in growth were considered as part of the phenotypic outcome." This is reasonable, but because standard length was reported only for fish at 90 dpf (not significantly different in any line), the reader is not given the opportunity to consider whether earlier differences in, e.g. surface area covered by osteoblasts at 14 dpf, could be accounted for by delayed or accelerated overall growth. Images in Figure S5 were not taken at the same magnification, further confounding this effort.

Comments on latest version:

The authors have largely addressed my comments by making changes to the text.

However, in response to one of my original comments ("It would be helpful to note the grouping of candidates into OI vs. BMD GWAS throughout the figures"), they added a sentence to this effect to the legends. However, because two of the lines were larval-lethal, the legends to Figs. S6-8 are now incorrect in referring to ten genes when only eight mutants are shown.

---

## [Referee Report · Reviewer #3 (Public review)]

The manuscript describes the use of CRISPR gene editing coupled with phenotyping mosaic zebrafish larvae to characterize functions of genes implicated in heritable fragile bone disorders (FBDs). Authors targeted six high-confident candidate genes implicated in severe recessive forms of FBDs and four Osteoporosis GWAS-implicated genes and observe varied developmental phenotypes across all crispants, in addition to adult skeletal phenotypes. While the study lacks insight on underlying mechanisms that contribute to disease phenotypes, a major strength of the paper is the streamlined method that produced significant phenotypes for all candidate genes tested. It also represents a significant increase in number of candidate genes tested using their crispant approach beyond the single mutant that was previously published.

One weakness was the variability of developmental phenotypes, addressed by authors in the Discussion. This might be a product of maternal transcripts not targeted by CRISPR or genetic compensation, which authors have not fully explored. Overall, the paper was well-written and easy to read.

Comments on latest version:

The authors have addressed many concerns in this revision. Figure 1 and Table 2 are much improved.

While details of orthologous gene expression profiles of target genes is a welcome addition, other features of target genes remain unaddressed. For example, do genes with maternally deposited transcript exhibit dampened phenotypes? Or does genetic compensation impact certain genes more than others? Since authors state that the study represents a methods paper, it will be important for users to understand the caveats of gene selection to effectively implement and interpret results of the approach.

---

## [Author Response]

The following is the authors’ response to the original reviews.

**Public Reviews:**

**Reviewer #1 (Public review):**
Summary:In this work, a screening platform is presented for rapid and cost-effective screening of candidate genes involved in Fragile Bone Disorders. The authors validate the approach of using crispants, generating FO mosaic mutants, to evaluate the function of specific target genes in this particular condition. The design of the guide RNAs is convincingly described, while the effectiveness of the method is evaluated to 60% to 92% of the respective target genes being presumably inactivated. Thus, injected F0 larvae can be directly used to investigate the consequences of this inactivation.Skeletal formation is then evaluated at 7dpf and 14dpf, first using a transgenic reporter line revealing fluorescent osteoblasts, and second using alizarin-red staining of mineralized structures. In general, it appears that the osteoblast-positive areas are more often affected in the crispants compared to the mineralized areas, an observation that appears to correlate with the observed reduced expression of bglap, a marker for mature osteoblasts, and the increased expression of col1a1a in more immature osteoblasts.Finally, the injected fish (except two lines that revealed high mortality) are also analyzed at 90dpf, using alizarin red staining and micro-CT analysis, revealing an increased incidence of skeletal deformities in the vertebral arches, fractures, as well as vertebral fusions and compressions for all crispants except those for daam2. Finally, the Tissue Mineral Density (TMD) as determined by micro-CT is proposed as an important marker for investigating genes involved in osteoporosis.Taken together, this manuscript is well presented, the data are clear and well analyzed, and the methods are well described. It makes a compelling case for using the crispant technology to screen the function of candidate genes in a specific condition, as shown here for bone disorders.Strengths:Strengths are the clever combination of existing technologies from different fields to build a screening platform. All the required methods are comprehe Zebrafish tanks_13062024nsively described.

We would like to thank the reviewer for highlighting the strengths of our paper.

Weaknesses:One may have wished to bring one or two of the crispants to the stage of bona fide mutants, to confirm the results of the screening, however, this is done for some of the tested genes as laid out in the discussion.

We thank the reviewer for their comment. We would like to point out that indeed similar phenotypes have been observed in existing models, as mentioned in the discussion section.

**Reviewer #2 (Public review):**
Summary:More and more genes and genetic loci are being linked to bone fragility disorders like osteoporosis and osteogenesis imperfecta through GWAS and clinical sequencing. In this study, the authors seek to develop a pipeline for validating these new candidate genes using crispant screening in zebrafish. Candidates were selected based on GWAS bone density evidence (4 genes) or linkage to OI cases plus some aspect of bone biology (6 genes). NGS was performed on embryos injected with different gRNAs/Cas9 to confirm high mutagenic efficacy and off-target cutting was verified to be low. Bone growth, mineralization, density, and gene expression levels were carefully measured and compared across crispants using a battery of assays at three different stages.Strengths:(1) The pipeline would be straightforward to replicate in other labs, and the study could thus make a real contribution towards resolving the major bottleneck of candidate gene validation.(2) The study is clearly written and extensively quantified.(3) The discussion attempts to place the phenotypes of different crispant lines into the context of what is already known about each gene's function.(4) There is added value in seeing the results for the different crispant lines side by side for each assay.

We would like to thank the reviewer for highlighting the strengths of our paper.

Weaknesses:(1) The study uses only well-established methods and is strategy-driven rather than question/hypothesis-driven.

We thank the reviewer for this correct remark. The mayor aim of this study was to establish a workflow for rapid *in vivo* functional screening of candidate genes across a broad range of FBDs.

(2) Some of the measurements are inadequately normalized and not as specific to bone as suggested:(a) The measurements of surface area covered by osteoblasts or mineralized bone (Figure 1) should be normalized to body size. The authors note that such measures provide "insight into the formation of new skeletal tissue during early development" and reflect "the quantity of osteoblasts within a given structure and [is] a measure of the formation of bone matrix." I agree in principle, but these measures are also secondarily impacted by the overall growth and health of the larva. The surface area data are normalized to the control but not to the size/length of each fish - the esr1 line in particular appears quite developmentally advanced in some of the images shown, which could easily explain the larger bone areas. The fact that the images in Figure S5 were not all taken at the same magnification further complicates this interpretation.

We thank the reviewer for this detailed and insightful remark. We agree with the reviewer and recognize that the results may be influenced by size differences. However, we do not normalize for size, as variations in growth were considered as part of the phenotypic outcome. This consideration has been addressed in the discussion section.

Line 335-338: ‘Although the measurements of osteoblast-positive and mineralized surface areas may be influenced by size differences among some of the crispants, normalization to size parameters was not conducted, as variations in growth were considered integral to the phenotypic outcome.’

Line 369: ‘Phenotypic variability in these zebrafish larvae can be attributed to several factors, including crispant mosaicism, allele heterogeneity, environmental factors, differences in genomic background **and development**, and slightly variable imaging positioning.’

(b) Some of the genes evaluated by RT-PCR in Figure 2 are expressed in other tissues in addition to bone (as are the candidate genes themselves); because whole-body samples were used for these assays, there is a nonzero possibility that observed changes may be rooted in other, non-skeletal cell types.

We thank the reviewer for this valuable comment. We acknowledge that the genes assessed by RT-PCR are expressed in other tissues beyond bone. This consideration has been addressed in the discussion section.

Line 362-365: “However, it is important to note that the genes evaluated by RT-PCR are not exclusively expressed in bone tissue. Since whole-body samples were used for expression analysis, there is a possibility that the observed changes in gene expression may be influenced by other non-skeletal cell types”.

(3) Though the assays evaluate bone development and quality at several levels, it is still difficult to synthesize all the results for a given gene into a coherent model of its requirement.

We appreciate the reviewer’s remark. We acknowledge that the results for the larval stages exhibit variability, making it challenging to synthesize them into a coherent model. However, it is important to emphasize that all adult crispant consistently display a skeletal phenotype. Consequently, the feasibility and reproducibility of this screening method are primarily focusing on the adult stages. This consideration has been addressed in the discussion section of the manuscript.

Line 391-399: ‘In adult crispants, the skeletal phenotype was generally more penetrant. All crispants showed malformed arches, a majority displayed vertebral fractures and fusions and some crispants exhibited distinct quantitative variations in vertebral body measurements. This confirmed the role of the selected genes in skeletal development and homeostasis and their involvement in skeletal disease and established the crispant approach as a valid approach for rapidly providing in vivo gene function data to support candidate gene identification.’

(4) Several additional caveats to crispant analyses are worth noting:(a) False negatives, i.e. individual fish may not carry many (or any!) mutant alleles. The crispant individuals used for most assays here were not directly genotyped, and no control appears to have been used to confirm successful injection. The authors therefore cannot rule out that some individuals were not, in fact, mutagenized at the loci of interest, potentially due to human error. While this doesn't invalidate the results, it is worth acknowledging the limitation.

We thank the reviewer for this valuable remark. We recognize the fact that working with crispants has certain limitations, including the possibility that some individuals may carry few or no mutant alleles. To address this issue, we use 10 individual crispants during the larval stage and 5 during the adult stage. Although some individuals may lack the mutant alleles, using multiple fish helps reduce the risk of false negatives.

Furthermore, we perform NGS analysis on pools of 10 embryos from the same injection clutch as the fish used in the various assays to assess the indel efficiency. While there remains a possibility of false negatives, the overall indel efficiency, as indicated by our NGS analysis, is high (>90%), thereby reducing the likelihood of having crispants with very low indel efficiency. We included this in the discussion.

Line 387-390: ‘While there remains a possibility of false negatives, the overall indel efficiency, as indicated by our NGS analysis, is high (>90%), thereby reducing the likelihood of having crispants with very low indel efficiency.’

(b) Many/most loci identified through GWAS are non-coding and not easily associated with a nearby gene. The authors should discuss whether their coding gene-focused pipeline could be applied in such cases and how that might work.

The authors thank the reviewer for this insightful comment. Our study is focused on strong candidate genes rather than non-coding variants. We recognize that the use of this workflow poses challenges for analyzing non-coding variants, which represents a limitation of the crispant approach. We have addressed this issue in the discussion section of the manuscript.

Line 131: ‘Gene-based’

Line 453: ‘Gene-based’

Line 311-314: ‘It is important to note that this study focused on candidate genes for osteoporosis, not on the role of specific variants identified in GWAS studies. Non-coding variants for instance, which are often identified in GWAS studies, present significant challenges in terms of functional validation and interpretation.’

**Reviewer #3 (Public review):**
Summary:The manuscript "Crispant analysis in zebrafish as a tool for rapid functional screening of disease-causing genes for bone fragility" describes the use of CRISPR gene editing coupled with phenotyping mosaic zebrafish larvae to characterize functions of genes implicated in heritable fragile bone disorders (FBDs). The authors targeted six high-confident candidate genes implicated in severe recessive forms of FBDs and four Osteoporosis GWAS-implicated genes and observed varied developmental phenotypes across all crispants, in addition to adult skeletal phenotypes.A major strength of the paper is the streamlined method that produced significant phenotypes for all candidate genes tested.

We would like to thank the reviewer for highlighting the strengths of our paper.

A major weakness is a lack of new insights into underlying mechanisms that may contribute to disease phenotypes, nor any clear commonalities across gene sets. This was most evident in the qRT-PCR analysis of select skeletal developmental genes, which all showed varied changes in fold and direction, but with little insight into the implications of the results.

We thank the reviewer for this insightful remark. We want to emphasize that this study focusses on establishing a new screening method for candidate genes involved in FBDs, rather than investigating the underlying mechanisms contributing to disease phenotypes. However, to investigate the underlying mechanisms in these crispants, the creation of bona fide mutants is necessary. We have included this consideration in the discussion.

Furthermore, we acknowledge that the results for the larval stages exhibit variability, which can complicate the interpretation of these findings. This is particularly true for the RT-PCR analysis, where whole-body samples were used, raising the possibility that other tissues may influence the expression results. Therefore, our primary focus is on the adult stages, as all crispants display a skeletal phenotype at this age. We have elaborated on this point in the discussion.

Line 462-463: ‘Moreover, to explore the underlying mechanisms contributing to disease phenotypes, it is essential to establish stable knockout mutants derived from the crispants’.

Line 391-399: ‘In adult crispants, the skeletal phenotype was generally more penetrant. All crispants showed malformed arches, a majority displayed vertebral fractures and fusions and some crispants exhibited distinct quantitative variations in vertebral body measurements. This confirmed the role of the selected genes in skeletal development and homeostasis and their involvement in skeletal disease and established the crispant approach as a valid approach for rapidly providing in vivo gene function data to support candidate gene identification.’

Ultimately, the authors were able to show their approach is capable of connecting candidate genes with perturbation of skeletal phenotypes. It was surprising that all four GWAS candidate genes (which presumably were lower confidence) also produced a result.

We appreciate the reviewer’s comment. We would like to direct attention to the discussion section, where we offer a possible explanation for the observation that all four GWAS candidate genes produce a skeletal phenotype.

Line 460-410: 'The more pronounced and earlier phenotypes in these zebrafish crispants are most likely attributed to the quasi knock-out state of the studied genes, while more common less impactful variants in the same genes result in typical late-onset osteoporosis (Laine et al., 2013) . This phenomenon is also observed in knock-out mouse models for these genes (Melville et al., 2014)(Coughlin et al., 2019).’

These authors have previously demonstrated that crispants recapitulate skeletal phenotypes of stable mutant lines for a single gene, somewhat reducing the novelty of the study.

We thank the reviewer for this comment and appreciate their concern. We have indeed demonstrated that crispants can recapitulate the skeletal phenotypes observed in stable mutant lines for the osteoporosis gene LRP5. However, we would like to highlight that the current study represents the first large-scale screening of candidate genes associated with bone disorders, including genes related to both OI and osteoporosis. We have included this information in both the abstract and the discussion

Line 60-62: ‘We advocate for a novel comprehensive approach that integrates various techniques and evaluates distinct skeletal and molecular profiles across different developmental and adult stages.’

Line 456-457: ‘While this work represents a pioneering effort in establishing a screening platform for skeletal diseases, it offers opportunities for future improvement and refinement.’

**Recommendations for the authors:**

**Reviewer #1 (Recommendations for the authors):**
(1) Figure 1a: what does the differential shading of the bone elements represent? Explain in the legend.

The differential shading doesn't represent anything specific. It's simply used to enhance the visual appeal and to help distinguish between the different structures. We removed the shading in the figure.

(2) Supplementary Figures 2-5: should the numbering of these figures be also in order of appearance in the text? I understand that the authors prefer to associate the transgenic and the alizarin red-stained specimens, however, the reading would be easier that way.

We changed this accordingly.

(3) Lines 275-276: "no significant differences in standard length (Figure 4a)": should be Figure 4b.

The suggested changes are incorporated in the manuscript.

Line 276-277: ‘Among the eight crispants that successfully matured into adulthood, none exhibited significant differences in standard length and head size (n=5 fish per crispant) (Figure 4b).’

(4) Line 277 "larger eye diameter": should be Figure 4b.

The suggested changes are incorporated in the manuscript.

Line 378: ‘However, esr1 crispants were observed to have notably larger eye diameters (Figure 4b).’

(5) Line 280: "no obvious abnormalities were detected (Figure 4b,c)": should be Figure 4a, c. Note that the authors may reconsider the a, b, c numbering in Figure 4 by inverting a and b.

The suggested changes are incorporated in the manuscript.

Line 278-281: ‘All these crispants demonstrated various abnormalities in the caudal part of the vertebral column such as fusions, compressions, fractures, or arch malformations, except for daam2 crispants where no obvious abnormalities were detected (Figure 4a,c; Supplementary Figure 6).’

(6) Table 2: This table, which recapitulates all the results presented in the manuscript, is in the end the centerpiece of the work. It is however difficult to read in its present form. Three suggestions:- Transpose it such that each gene has its own column, and the lines give the results for the different measurements- Place the measurements that result in "ns" for all crispants at the end (bottom) of the table.- Maybe bring the measurements at 7dpf, 14dpf, and 90 dpf together.

We agree with the reviewer and have added a new table where we transposed the data. However, we chose not to place the measurements that resulted in 'ns' for all crispants at the end of the table, as we believe it is important to track the evolution of the phenotype over time. Where possible, we have grouped the measurements for 7 dpf and 14 dpf together.

**Reviewer #2 (Recommendations for the authors):**
(1) It would help to justify why these particular area measurements are appropriate for this set of candidate genes, which were selected based on putative links to bone quality rather than bone development.

The selected methods are among the most commonly used to evaluate bone phenotypes. They are straightforward to reproduce, as well as cost- and time-effective. The strength of this approach lies in its use of simple, reproducible techniques that form the foundation for characterizing bone development. Although the candidate genes were chosen based on their putative links to bone quality, early skeletal phenotypes can already be observed during bone development.

The mineralized surface area of the total head and specific head structures was selected to evaluate the degree of mineralization in early skeletal development, as mineralization is a direct indicator of bone formation. Additionally, the osteoblast-positive surface areas were measured to provide insight into the formation of new skeletal tissue during early development. Osteoblasts, as active bone-forming cells, are essential for understanding bone growth and the dynamics of skeletal phenotypes.

Examples in the manuscript:

Line 212-214: ‘The osteoblast-positive areas in both the total head and the opercle were then quantified to gain insight into the formation of new skeletal tissue during early development.’

Line 221-223: ‘Subsequently, Alizarin Red S (ARS) staining was conducted on the same 7 and 14 dpf crispant zebrafish larvae in order to evaluate the degree of mineralization in the early skeletal structures.’

(2) Reword: The opercle bone is the earliest forming bone of the opercular series, and appears to be what the authors are referring to as the "operculum" at 7-14 dpf. The operculum is the larger structure (gill cover) in which the opercle is embedded. It would be more accurate to simply refer to the opercle at these stages.

We agree with this comment and changed the text accordingly.

(3) Define BMD and TMD at first usage.

BMD and TMD are now defined in the manuscript.

Line 41-43: ‘Six genes associated with severe recessive forms of Osteogenesis Imperfecta (OI) and four genes associated with bone mineral density (BMD), a key osteoporosis indicator, identified through genome-wide association studies (GWAS) were selected.’

Line 286-288: ‘For each of the vertebral centra, the length, tissue mineral density (TMD), volume, and thickness were determined and tested for statistical differences between groups using a regression-based statistical test (Supplementary Figure 7).’

(4) It would be helpful to note the grouping of candidates into OI vs. BMD GWAS throughout the figures.

We agree with this comment and added this to all figure legends.

‘The first four genes are associated with the pathogenesis of osteoporosis, while the last six are linked to osteogenesis imperfecta’

**Reviewer #3 (Recommendations for the authors):**
Major points:(1) For the Results, it would be useful to the Reader to justify the selection of human candidate genes and their associated zebrafish orthologs to model skeletal functions. For example, what are variants identified from human studies, and do they impact functional domains? Are these domains and/or proteins conserved between humans/zebrafish? Is there evidence of skeletal expression in humans/zebrafish?

Supplementary Table 4 lists the selected human candidate genes with reported mutations and/or polymorphisms associated with both skeletal and non-skeletal phenotypes. The table also includes additional findings from studies in mice and zebrafish. An extra column was now added to indicate gene conservation between human and zebrafish. We consulted UniProt (https://www.uniprot.org) and ZFIN (https://zfin.org) to assess the skeletal expression of these genes in human and zebrafish. All genes showed expression in the trabecular bone and/or bone marrow in humans, as well as in bone elements in zebrafish. We added this in the discussion.

Line 309: ‘All selected genes show skeletal expression in both human and zebrafish.’

Supplemental table 4 legend: ‘The conservation between human and zebrafish is reported in the last column.’

As part of this, some version of Supplementary Table 4 might be included as a main display to introduce the targeted genes, ideally separated by rare (recessive OI) vs. common disease (osteoporosis). In the case of common disease and GWAS hits, how did authors narrow in on candidate genes (which often have Mbp-scale associated regions spanning multiple genes)? Further, what is the evidence that the mechanism of action of the GWAS variant is haploinsufficiency modeled by their crispant zebrafish?

We have kept Supplementary Table 4 in the supplementary material but have referred to it earlier in the manuscript’s introduction. Consequently, the table has been renumbered from ‘Supplementary Table 4’ to ‘Supplementary Table 1’.

The selection of genes potentially involved in the pathogenesis of osteoporosis is based on the data from the GWAS catalog, which annotates SNPs using the Ensemble mapping pipeline. The available annotation on their online search interface includes any Ensemble genes to which a SNP maps, or the closest upstream and downstream gene within a 50kb window. Four genes were selected for this screening method based on the criteria outlined in the results section. In this study, we aim to evaluate the general involvement of specific genes in bone metabolism, rather than to model a specific variant.

Line 135-136 and 309-311: ‘An overview of the selected genes with observed mutant phenotypes in human, mice and zebrafish is provided in Supplementary Table 1.’

(2) Using the crispant approach does not impact maternally-deposited RNAs that would dampen early developmental phenotypes. Considering the higher variability in larval phenotypes, perhaps the maternal effect plays a role. The authors might investigate developmental expression profiles of their genes using existing RNA-seq datasets such as from White et al (doi: 10.7554/eLife.30860).

We thank the reviewer for this comment and agree with the possibility that maternally-deposited RNAs might have an impact on early developmental phenotypes. We included this in the discussion.

Line 369-372: ‘Phenotypic variability in these zebrafish larvae can be attributed to several factors, including crispant mosaicism, allele heterogeneity, environmental factors, differences in genomic background and development, maternally-deposited RNAs, and slightly variable imaging positioning.’

(3) While making comparisons within a clutch of mutant vs scrambled control is crucial, it is also important to ensure phenotypes are not specific to a single clutch. Do phenotypes remain consistent across different crosses/clutches?

Yes, phenotypes remain consistent across different crosses and clutches. We included images from a second clutch in the Supplementary material (Supplementary Figure 8) and refereed to it in the discussion.

Line 394-397: ‘Additionally, these skeletal malformations were consistently observed in a second clutch of crispants (Supplementary Figure 8), underscoring the reproducibility of these phenotypic features across independent clutches.’

(4) Understanding that antibodies may not exist for many of the selected genes for zebrafish, authors should verify haploinsufficiency using an RT-qPCR of targeted genes in crispants vs. controls.

We appreciate the reviewer’s suggestion to use RT-qPCR to examine expression levels of the targeted genes in crispants. However, previous experience suggests that relying on RNA expression to verify haploinsufficiency in zebrafish can be challenging. In zebrafish KO mutants, RT-qPCR often still detects gene transcripts, potentially due to incomplete nonsense-mediated decay (NMD) of the mutated mRNA, which may allow residual expression even in the absence of functional protein. As a more definitive approach, we prefer to use antibodies to confirm haploinsufficiency at the protein level. However, as the reviewer noted, generating and applying specific antibodies in zebrafish remains challenging.

(5) Please indicate how parametric vs. non-parametric statistical tests were selected for datasets.

We initially selected the parametric unpaired t-test, assuming the data were normally distributed with similar variances between groups. We verified the assumption of equal variances using the F-test, which was not significant across all assays. However, we did not assess the normality of the data directly, meaning we cannot confirm the normality assumption required for the t-test. Given this, we have opted to use the non-parametric Mann-Whitney U test, which does not require assumptions of normality, to ensure the robustness of our statistical analyses. We changed the Figures, the figure legends and the text accordingly.

(6) In the figures and tables, I recommend adding notation showing the grouping of the first four genes as GWAS osteoporosis, the next three genes as osteoblast differentiation, the next two genes as bone mineralization, and the final gene as collagen transport to orient the reader. One might expect there to be a clustering of phenotypic outcomes based on the selection of genes, and it would be easier to follow this. This would be particularly useful to include in Table 2.

Our primary objective is to assess the feasibility and reproducibility of the crispant screen rather than performing an in-depth pathway analysis or categorizing genes by biological processes. For this purpose, we have organized candidate genes based on their relevance to osteoporosis and Osteogenesis Imperfecta, without subdividing them further. We have clarified this focus in the figure legends, as suggested in an earlier recommendation.

(7) For Figure 1, consider adding a smaller zoomed version of 1a embedded in each sub-figure with each measured element highlighted to improve readability.

We agree with this comment and changed the figure accordingly.

Minor points:(1) Table 2 could be simplified to improve readability. The headers have redundancies across columns with varied time points and could be merged.

The suggested changes are incorporated in the manuscript (see earlier comment about this).

(2) "BMD" is not defined in the Abstract. This is a personal preference, but there were numerous abbreviations in the text that made it difficult to follow at times.

The suggested changes are incorporated in the manuscript (see earlier comment about this).